



# Detection and long-term quantification of methane emissions from an active landfill

Pramod Kumar[1], Christopher Caldow[1,2], Grégoire Broquet[1], Adil Shah[1], Olivier Laurent[1], Camille Yver-Kwok[1], Sebastien Ars[3], Sara Defratyka[1,6], Susan W. Gichuki[1], Luc Lienhardt[1], Mathis Lozano[1], Jean-Daniel Paris[1], Felix Vogel[3], Caroline Bouchet[4], Elisa Allegrini[4], Robert Kelly[4], Catherine Juery[5], Philippe Ciais[1]

[1]Laboratoire des Sciences du Climat et de l'Environnement (LSCE/IPSL), CEA-CNRS-UVSQ, Université Paris-Saclay, 91191 Gif-sur-Yvette, France
[2]Climate Science Centre, CSIRO Oceans and Atmosphere, Aspendale, VIC, 3195, Australia
[3]Climate Research Division, Environment and Climate Change Canada, Toronto M3H 5T4, Ontario, Canada
[4]SUEZ-Smart & Environmental Solutions, Tour CB21/16 place de l'Iris, 92040, La Défense, France
[5]TotalEnergies Laboratoire Qualité de l'Air (LQA), 69360 Solaize Cedex, France
[6]now at University of Edinburgh, United Kingdom and National Physical Laboratory, United Kingdom

*Correspondence to*: Pramod Kumar (pramod.kumar@lsce.ipsl.fr)

## Abstract

Landfills are a significant source of fugitive methane ($CH_4$) emissions which should be precisely and regularly monitored to reduce and mitigate net greenhouse gas emissions. In this study, we present long-term *in-situ* near-surface mobile atmospheric $CH_4$ mole fraction measurements (complemented by meteorological measurements from a fixed station) from 21 campaigns that cover approximately four-years from September 2016 to December 2020. These campaigns were utilized to regularly quantify the total $CH_4$ emissions from an active landfill in France. We use a simple atmospheric inversion approach based on a Gaussian plume dispersion model to derive $CH_4$ emissions. Together with the measurements near the soil surface mainly dedicated to the identification of sources within the landfill, measurements of $CH_4$ made on the landfill perimeter (near-field) helped us to provide some qualitative insights about the respective weight of the main areas of emissions. However, we hardly managed to extract a signal representative of the overall landfill emissions from these measurements, which limited our ability to derive robust estimates of the emissions when assimilating them in the atmospheric inversions. The analysis shows that the inversions based on the measurements from a remote road further away from the landfill (far-field) yielded more reliable estimates. According to these estimates, the total $CH_4$ emissions have a large temporal variability and range from ~0.4 t $CH_4$/d to ~7 t $CH_4$/d, with an average value of ~2.1 t $CH_4$/d. We find a weak negative correlation between these estimates of the $CH_4$ emissions and atmospheric pressure for the active landfill. However, this weak emission-pressure relationship is based on a relatively small sample of reliable emission estimates with large sampling gaps. More frequent robust estimations are required to better understand this relationship for an active landfill.

## 1 Introduction

Methane ($CH_4$) is Earth's second most important anthropogenic greenhouse gas after carbon dioxide (Hartmann et al., 2013; Kirschke et al., 2013), and has a much larger global warming potential (Etminan et al., 2016). $CH_4$ emissions are increasing (Jackson et al., 2020), resulting in a high growth rate of global annual average $CH_4$ mole fractions in the atmosphere reaching up to 1911.88±0.59 parts per billion (ppb) for 2022, more than two-and-a-half times preindustrial levels (Lan et al., 2022; Nisbet et al., 2020), despite a temporary pause between 1998 and 2007 (Bousquet et al., 2006; Rigby et al., 2008; Turner et al., 2019). According to the values reported by NOAA, the annual increases in 2020 (15.20±0.41 ppb) and 2021 (17.75±0.47 ppb) are the greatest observed since the systematic record began in 1983 (Lan et al., 2022). $CH_4$ is a short-lived radiative forcer and reducing its emissions will deliver an immediate reduction of net global warming. Fossil fuel extraction, agriculture, and waste



management are responsible for over half of all $CH_4$ emissions (Saunois et al., 2016). Reducing these anthropogenic emissions, as pledged in Glasgow by more than 100 countries
(https://www.globalmethanepledge.org/), is viewed as an effective wedge to meet the short-term objectives of the Paris Agreement, even though achieving long-term neutrality goals will require reducing carbon dioxide emissions as well.

Reducing fugitive emissions from landfills can make a valuable contribution to the Glasgow methane pledge (Dreyfus et al., 2022; Nisbet et al., 2020; Shindell et al., 2012), and the
European Union (EU) is planning on targets and regulations for this sector (European Union Methane Action Plan, 2022). Methane is produced in landfills during the anaerobic microbial decomposition of organic waste (Bingemer and Crutzen, 1987). Total waste emissions have increased in past decades (Jackson et al., 2020) roughly doubling between 1970 and 2010 (Fischedick et al., 2014). Landfills and waste constituted ~18% of total anthropogenic $CH_4$
emissions in the year 2017 (Jackson et al., 2020; Saunois et al., 2020). Society's reliance on landfills to store waste is set to increase with population growth and development (Hein et al., 1997; Hong et al., 2017; Lando et al., 2017). In the EU, anaerobic decomposition in the waste sector is the second largest methane source, accounting for ~18% of total emissions in the year 2018 (European Environment Agency, 2020, p.73). Waste management is nevertheless
regulated in the EU (Bourn et al., 2019; Daugela et al., 2020; Fjelsted et al., 2019; Scheutz et al., 2009)  and net land waste disposal emissions decreased by 46% between 1990 and 2018 (European Environment Agency, 2020, p.794) primarily through diverting organic waste away from storage in landfills (European Commission, 2020). Landfill emission mitigation is gaining traction (Bogner et al., 2008; Mønster et al., 2019), by curtailing organic waste reaching landfill
(Shams et al., 2017) and by recuperating the methane produced on-site as biogas (Duan et al., 2021; Scheutz et al., 2009). Although landfill biogas can be flared (Tratt et al., 2014), biogas collection and use for heat and electricity production is more and more implemented  (Bogner et al., 1995; Riddick et al., 2018; Themelis and Ulloa, 2007).

$CH_4$ flux estimates at the scale of individual sites have proven to be indispensable in the
establishment of effective landfill emissions regulation (Bogner and Matthews, 2003; Scheutz et al., 2009; Tratt et al., 2014). Bottom-up inventories of methane emissions can be derived from waste quantity, waste composition, and emission factors (Jha et al., 2008; Shams et al., 2017). But those inventory estimates can be far from accurate, as they rely on default emission factors that may not be representative of the real conditions on-site (Krautwurst et al., 2017;
Nisbet et al., 2019). Therefore, independent measurement-based flux estimates are vital to derive relevant values for individual sites and for the development of inventories which could reflect the high diversity of site-level management practices, technologies, and environmental conditions (Bourn et al., 2019; Cambaliza et al., 2015; Nisbet et al., 2020).

Estimating the $CH_4$ emissions of a landfill site based on on-site measurements can be
challenging. Landfills are spatially complex, with heterogeneous sources including point-scale and area-scale emission sources that can vary substantially over time (Fjelsted et al., 2019; Lando et al., 2017; Rachor et al., 2013). Depending on the flux quantification strategy, a knowledge of the spatial distribution of the sources within a site has been shown to be critical for effective emission quantification (Daugela et al., 2020; Riddick et al., 2018; Zazzeri et al.,
2015). Landfill emissions occur from both active (uncovered) and covered cells (Sonderfeld et al., 2017), as well as from infrastructure including pipes, wells, leachate ponds, and gas recuperation/processing facilities (Allen et al., 2019; Bogner et al., 1995; Emran et al., 2017). This surface heterogeneity means that emission quantification methods must be adapted to the configuration of each site (Bourn et al., 2019; Mønster et al., 2019). For example, flux
chambers deliver precise surface fluxes of very local emissions at the scale of about 1 m$^2$



(Fjelsted et al., 2019; Jha et al., 2008; Lando et al., 2017), but require a sufficient spatial sampling density for adequate site characterization. Manual chamber installation and maintenance can be arduous.

Alternatively, atmospheric inversion techniques can be employed to quantify fluxes. The computation of emissions from landfills with such techniques often relies on measurements of the methane mole fractions downwind to the sites (Allen et al., 2019; Ars et al., 2017; Lohila et al., 2007; Mønster et al., 2019). These measurements can be utilized in mass balance modelling, tracer release methods, or inverse atmospheric dispersion models to quantify landfill methane fluxes (Ars et al., 2017; Duan et al., 2022; Foster-Wittig et al., 2015; Krautwurst et al., 2017; Riddick et al., 2018; Sonderfeld et al., 2017; Yacovitch et al., 2018). This approach can capture emissions from a large area of the landfill, from multiple area and/or point sources, or from the entire site (Bourn et al., 2019).

Several platforms can be used to sample the atmospheric methane mole fractions within and around a landfill, each with advantages and disadvantages. Examples include stationary towers (Riddick et al., 2018), satellites (Maasakkers et al., 2022; Tu et al., 2022), manned aircraft (Cambaliza et al., 2015; Gasbarra et al., 2019; Krautwurst et al., 2017; Tratt et al., 2014), unmanned aerial vehicles (UAV) (Allen et al., 2019; Bel Hadj Ali et al., 2020), and a mobile ground-based laboratory (MGL) performing mobile plume transects at ground level (Ars et al., 2017; Foster-Wittig et al., 2015; Sonderfeld et al., 2017). Satellites can provide broad spatiotemporal coverage and resolution to monitor individual landfill methane emissions; however, they are only applicable to strongly emitting landfills with total emissions on an order of $1 \, t \, CH_4 \, h^{-1}$ due to their detection limit using the currently available measurement technology (Maasakkers et al., 2022; Tu et al., 2022). Aerial aircraft or UAV $CH_4$ mole fraction measurements, with wind measurements, have great potential in monitoring landfill emissions (Allen et al., 2019; Gasbarra et al., 2019). However, UAVs and aircraft cannot sample for prolonged periods, providing only a snapshot of emission flux (Mønster et al., 2019). Continuous atmospheric measurements from stationary *in situ* and precise sensors located within or close to a site can provide long-term monitoring of emissions with a much lower detection limit (Kumar et al., 2022; Riddick et al., 2018). However, the deployment of a dense network of sensors is limited by cost, more specifically, by the lack of precise and reliable low-cost $CH_4$ sensors (Fox et al., 2019; Mønster et al., 2019). The mobile ground-based laboratory (MGL) measurements can be used for routine sampling of the total emissions from a landfill, throughout its life-cycle. MGLs are typically equipped with a satellite positioning module, gas analyzers, and wind sensors. MGLs can provide transects of the plumes from landfills with both high spatial resolution and coverage, e.g. by driving on a nearby downwind sampling road (Kumar et al., 2022, 2021; Scheutz et al., 2011; Zazzeri et al., 2015). They can also provide some insight into the location of potential emission sources when sampling near the source and combining sampled mole fractions with wind measurements (Ars et al., 2020). If focusing on a single site and planning campaigns under favorable wind conditions, they can support routine analysis of a site's methane emissions. However, MGL operation can be labor intensive and sampling can be limited to road infrastructure and favorable winds for adequate downwind positioning. In addition to MGL sampling of downwind landfill methane plumes, a tracer gas may be released at a known rate near to a targeted source to estimate methane fluxes by exploiting mole fraction ratios between methane and the tracer gas (Czepiel et al., 1996; Scheutz et al., 2011; Yver Kwok et al., 2015). In this study, we conducted MGL measurements to analyze methane emissions from an active landfill.





The main objective of this study is to analyze methane emissions from an active landfill site near Paris, France over a prolonged period of approximately four years, between September 2016 and December 2020 during which it highly evolved. The studied landfill is an ~0.18 km² managed landfill site, operated by SUEZ, and it has been in operation since 2005. The site is composed of several cells, some being covered by membranes, where biogas is recuperated from a network of wells connected to pipes, and some being openly exposed to air while being filled with waste. In this study, we use a simple inverse atmospheric dispersion modelling approach to quantify $CH_4$ emissions using downwind near-surface mobile $CH_4$ mole fraction measurements complemented by meteorological measurements from a fixed station, for 21 MGL campaigns. These MGL campaigns were undertaken within the framework of various projects (mainly TRACE, but also, initially, wastemiti and bridGES) in collaboration with SUEZ (Ars, 2017; Ars et al., 2017; Vogel, 2016) and were conducted mainly in three phases: September 2016 to December 2016, August 2017 to October 2017, and July 2018 to December 2020. We regularly quantify the net methane emissions of the site and their evolution over time. We also provide some information on specific sources within the site using near-site transects combined with complementary on-foot targeted leak detection (henceforth referred to as "sniffing") measurements, and on emissions spatial distribution through inversions using near-site transects.

Our analysis of the data for the methane emissions is based on a simple Gaussian plume model which is driven by on-site meteorological measurements and has been utilized and evaluated previously for the inversions of methane emissions from controlled release experiments (Kumar et al., 2022, 2021). In Section 2, we describe the site and our data collection. Section 3 presents a first attempt at deriving information on the distribution of the emissions within the landfill based on the measurement from the foot sniffing and from the MGL transects close to the site. We describe our inversion approach in Section 4 followed by the results and discussions respectively in Sections 5 and 6 and our conclusions in Section 7.

## 2 Materials and methods

### 2.1 Site description

The studied landfill is located about 35 km south-east of Paris (longitude: 2° 44.381'E, latitude: 48° 38.434'N, area: ~0.18 km², altitude above the sea level: ~100 to 120 m; Figure 1). It is close (about 200-300 m east) to an older closed landfill (1974-2004), which has been completely covered since 2005. The studied landfill began receiving waste in 2005 with its last waste received in 2022. It has an overall waste capacity of ~3.05 Mt. By the end of 2020, it had received approximately 97% of this capacity. The landfill has been divided up into approximately six cells, each being progressively filled and compacted before being covered with a non-permeable membrane overladen with 0.8 m of soil. The site is equipped with a leachate and biogas collection network to collect and treat biogas and leachate to be used on site. Two gas engines are installed on site to generate electricity with the landfill gas. The cells of the landfill have been filled in a counter-clockwise fashion starting with the NE corner and progressing around to the SE corner where waste reception was ongoing during this study (see Figure S1.1 in the supplementary information (SI) SI-1). Waste is deposited and compacted during operational hours which are 07:00 to 15:00 (local time) during weekdays. At the end of



each day, the active area of the landfill is covered with clay or soil in order to minimize odor
and biogas emissions as well as animal activity overnight.

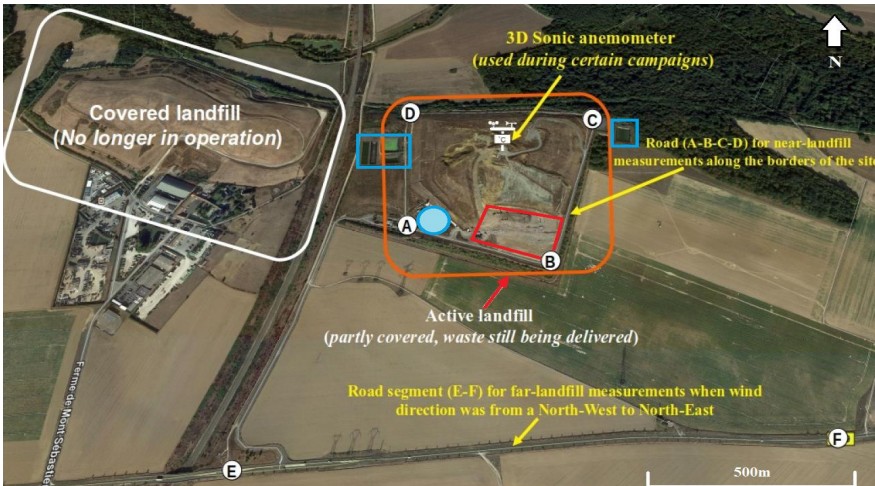

**Figure 1:** Satellite image (source: Google Earth, © Google Earth) of the studied landfill (orange rectangle on the right side of the figure), an older closed landfill (white rectangle on the left), and its surrounding area. The red quadrangle, blue rectangles, and blue circle designate the locations of the active landfill cell during the period 2018-2020, leachate ponds, and biogas valorization plant, respectively. The letters A to F designate the ends of the segments of the roads along which the mobile measurements were taken during the field measurements. Most of the measurements were taken along the road segments A to B and B to C close to the landfill, or along the E to F "remote roads", which is henceforth referred to as "EF" (E and F refer to the end-points of any distant sampling road; however, the measurements only from EF "remote roads" south of the landfill were used for inversions). During most of the campaigns, a 3-D sonic anemometer was installed at an elevated location near the center of the landfill.

The topography of the landfill is complex. It may be generally described as a hill that rises towards the center and slopes away towards the edge. The highest point of the landfill is a few tens of meters above the outer edges with variations in time due to the evolution of the landfill.
The area surrounding the landfill is generally flat as it has been used as cropland. The closed landfill exhibits similar topography to the studied one with a similar height and a slightly greater extent (area ~0.25 $km^2$; Figure 1). Based on measurement surveys conducted previously (Vogel, 2016) and during this study, we see that there is no significant $CH_4$ signal from this closed landfill in our measurements targeting the active landfill.

**2.2 Scientific instrumentation**

**2.2.1 Mobile Ground Laboratory and sniffing measurement framework**

Atmospheric sampling was performed within and around the studied landfill using an MGL. A vehicle was equipped with 1 to 3 gas analyzers that continuously measured in-situ $CH_4$ mole fraction, the mole fraction of additional trace gases ($CO_2$, CO, $C_2H_2$, $H_2O$), and isotope mole
fractions ($\delta^{13}C$ in $CH_4$ and $\delta^{13}C$ in $CO_2$) depending on the type of analyzer. We utilized a variety of high-precision cavity enhanced absorption spectroscopy gas analyzers: the Picarro G2203 ($CH_4$, $C_2H_2$, $H_2O$), G2401 ($CH_4$, $CO_2$, CO, $H_2O$), and G2201-*i* ($CH_4$, $CO_2$, $\delta^{13}C$ in $CH_4$ and $\delta^{13}C$ in $CO_2$), which use cavity ring down spectrometry, the ABB Ultra-portable Greenhouse Gas Analyzer (ABB-UGGA) and Micro-portable Greenhouse Gas Analyzer (ABB-MGGA),
which use off-axis integrated cavity output spectroscopy, and a LI-COR LI-7810 prototype gas analyzer, which uses optical feedback-cavity enhanced absorption spectroscopy (see Table 1).





The accuracy of all gas analyzers was verified in the laboratory using low (1.98 ± 0.11 (1σ) ppm) and high (6.14 ± 0.23 (1σ) ppm) $CH_4$ mole fractions calibration standards that are traceable to World Meteorological Organization (WMO) greenhouse gas scales
(WMOX2004A; WMO GAW report No. 255).

The gas analyzers were connected to an air inlet located towards the front of the MGL roof, using ¼ inch Synflex 1300 tubing. Power was supplied by gel lead-acid batteries (12 V, 150 Ah) with either one battery connected directly to a power inverter (12 V / 230 V, DC / AC) or two batteries connected in series (24 V / 230 V, DC / AC). A Global Positioning Satellite (GPS)
module inside the MGL recorded the sampling position at 1 Hz during the campaigns. All measurements were synchronized to UTC. Moreover, the net gas analyzer time response (including the delay induced by the sampling line) was initially determined on-site by providing a short burst of breath into the air inlet and then timing the response, for post correction of the campaign data set

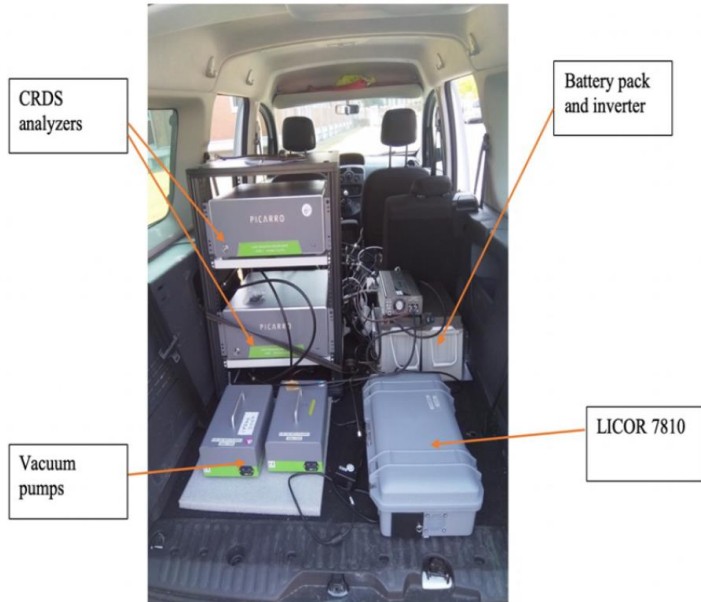


**Figure 2:** Example of the mobile instrument configuration as setup in a vehicle. Different combinations of instruments were used for the different campaigns, as detailed in Table 1. The LICOR™ 7810 in the picture was on loan to LSCE and was used in one campaign only.

### 2.2.2 Meteorological measurements

Reliable meteorological and micrometeorological measurements are required to support the analysis of the gas mole fraction measurements, and in particular to characterize atmospheric conditions in the Gaussian plume dispersion model used for the inversion modeling to estimate $CH_4$ emissions from the landfill. For the MGL measurement campaigns during 2016-2017, a 2-dimensional (2-D) anemometer meteorological station, measuring one-minute averaged
wind speed and direction, was permanently installed at ~10 m height above ground level (agl) near the biogas valorization plant (Figure 1). For the majority of campaigns between 2018 and 2020, a 3-dimensional (3-D) sonic anemometer (Gill Instruments WindMaster 3-Axis Anemometer) was installed near the center and the highest point of the landfill where nearby obstacles were limited. The anemometer was installed on a mast at a height of between about



2 to 7 m agl. Data from the 3-D sonic anemometer was recorded at 20 Hz using a Raspberry Pi 3B+ logging computer. For 4 of the 21 campaigns documented in this study, wind measurements were not made on site and therefore computations relied on wind observation data from the nearby Melun meteorological station (48°36'37" N, 2°40'46" E) operated by Meteo France, which is located ~5.5 km SW of the studied landfill. For all campaigns, we used

atmospheric pressure, air temperature, and humidity measurements from the Melun station.

## 2.3   Measurement strategy

The monitoring of the $CH_4$ emissions from the landfill site posed two major challenges related to spatiotemporal variability: a) that of the identification of the different methane sources on the site, which can either be very localized (hotspots) or more diffuse sources, and b) that of

the estimation of their emissions which can vary over time due to changing operational or external parameters e.g., atmospheric conditions. In order to tackle these challenges, the main strategy for our measurement campaigns was to (a) continuously measure $CH_4$ mole fractions across the atmospheric plumes downwind of the landfill (obtaining "plume cross-sections") during MGL surveys of at least one hour along roads close to and distant from the site, and (b)

to conduct some on-foot "sniffing" within the landfill to identify local methane hotspots and to characterize the potential emissions sources. The longer-term (seasonal and interannual) temporal variability was also addressed by conducting campaigns over several years. MGL campaigns were performed on the road along the perimeter of the site between points A, B, C, and D in Figure 1, and/or along the EF "remote roads", where E and F refer to the end-points

of any distant sampling road. Due to accessibility limitations of suitable EF "remote roads", the campaigns generally targeted days when winds were from the north-west to the north-east to ensure that the mobile transects on the EF "remote roads" south to the landfill lay downwind of the site and would intersect the landfill $CH_4$ emission plume. The measurements conducted on these southern EF "remote roads" (subsequently referred to as "EF roads") are primarily

used for inversions. When planning the campaigns, such suitable meteorological conditions were chosen from weather forecasts at least a day in advance. All campaigns were carried out between mid-morning and early afternoon on weekdays when the site could be accessed.

## 2.4   General information on the campaigns

We conducted a total of 27 MGL campaigns between September 2016 and December 2020

with an average period of revisit of ~42 days, ranging between 7 and 149 days. However, the measurements made during six MGL campaigns are excluded from the study because, during these campaigns, the GPS MGL position was not recorded which prevented us from conducting robust analysis. Therefore, we conducted our analysis for the 21 campaigns listed in Table 1. During two of these MGL campaigns (August 29, 2019 and March 04, 2020), we

simultaneously conducted additional foot-based sniffing measurements at the ground level within and around the site, for locating specific point or area sources within the landfill site. In both "sniffing" campaigns, a portable ABB MGGA was used to measure $CH_4$ mole fractions, with a GPS positioning module, whilst walking around suspected hotspots within the landfill. We obtained an average of about 10 plume cross-sections per campaign. For 11 of these 21

MGL campaigns, we have plume cross-sections on EF roads which were used in the inverse modelling framework (Section 4) for the estimation of the total methane emissions from the landfill. Sampling was performed along the ABCD road (see Figure 1) in all but one of the 21 MGL campaigns, under a variety of different wind conditions. This sampling aimed to provide insight into the spatial distribution of emissions within the landfill, but we also expected that

plume cross-sections along these roads could support the inversion of the total emission from the landfills or from some of its main areas of emissions, in particular from its different cells.





Table 1 summarizes information on the gas analyzers used, the number of ABCD and/or EF plume cross-sections conducted, and the meteorological and/or turbulence parameters for all the selected 21 campaigns. For each selected campaign, Figures S1.2 to S1.22 in the SI-1 show

the $CH_4$ mole fraction time series, plume cross-sections, and the corresponding wind conditions according to on-site meteorological measurements or to local wind conditions in four campaigns from the Melun weather station. The wind speed ($U$) and wind direction ($\theta$) for each campaign are averaged over each campaign period. The averaged wind speeds in all of the selected campaigns varied from ~1 ms$^{-1}$ to ~7 ms$^{-1}$ (Table 1). During two of the 21 campaigns,

averaged wind speeds were equal or below 1.5 ms$^{-1}$. The use of a Gaussian plume model for such low wind speed conditions leads to higher uncertainty in $CH_4$ emission estimates (Kumar et al., 2022, 2021). However, during these campaigns, we had $CH_4$ measurements along the EF road that appeared to be suitable and we still attempted inversions to estimate the $CH_4$ emissions from the site with these measurements. During several campaigns, $CH_4$ mole fraction

measurements were made even when unfavorable winds were coming from the east to the southwest (Table 1). The mobile transects in these campaigns were mostly conducted along the ABCD road and/or along a westside EF "remote road", very near to the landfill. In other campaigns, the wind directions ranged between the north-west to north-east directions which enabled us to use MGL sampling on both the ABCD and EF roads (Table 1).

Whenever the high-frequency data from the 3-D sonic was available, the essential turbulence parameters, the Obukhov length ($L$), surface friction velocity ($u_*$), and standard deviation of wind velocity fluctuations ($\sigma_u$, $\sigma_v$, $\sigma_w$) were computed over each campaign period. All of the campaigns were conducted during daytime and thus, for the campaigns with 3-D sonic data, the negative sign and magnitude of the Monin-Obukhov stability parameter (1/$L$) indicate that

the atmospheric stability varied from near-neutral to unstable and very unstable conditions. For the remaining campaigns, the Pasquill-Gifford-Turner (PGT) atmospheric stability classes, characterized based on the wind measurements (Turner, 1970), varied from neutral (PGT class: D) to very unstable (PGT class: A) conditions.

The background $CH_4$ mole fraction outside the plume cross-sections conducted along ABCD

and EF roads in each campaign was taken as the first percentile of the $CH_4$ measurements, so that the enhancements in $CH_4$ due to landfill emissions could be determined from this background. Measurements obtained upwind of the landfill, usually between points C-D (Figure 1), confirmed that using the first percentile was appropriate to characterize the background $CH_4$ field on top of which lies the plumes from the landfill. This approach of

deriving a background from field measurements, eliminates any potential offset issues in the gas analyzers, thereby reducing instrumental uncertainty.

Across all the 21 selected campaigns, the maximum $CH_4$ enhancement above the background reached up to ~70 ppm and ~3.5 ppm, for the ABCD and EF roads, respectively. We computed averages of the $CH_4$ mole fraction enhancements for segments of these roads from the different

mobile transects along them. To compute these averages, a road was divided into equidistant segments with an averaged distance interval between the measurement locations. Then, the $CH_4$ mole fractions at each segment point were averaged by using the nearest point $CH_4$ mole fraction values from different mobile transects. These averaged $CH_4$ mole fractions are shown in Figure 3 for the EF roads and in SI figures S1.2-S1.22 for all the roads. During most of the

campaigns when the wind was coming from the north-west to north-east direction, the high $CH_4$ mole fraction enhancements either represented individual plume cross-sections or averaged $CH_4$ plumes, which were observed along the road segments A-B or B-C (Figures S1.2 to S1.22). The averaged $CH_4$ plumes from different campaigns along the ABCD road systematically show multiple $CH_4$ peaks nearly at the same downwind locations during the



series of MGL cross-sections. These different $CH_4$ peaks indicate the heterogeneous distribution of $CH_4$ emissions within the landfill. The different plume cross-sections and corresponding averaged $CH_4$ plume along the EF road show a more unimodal plume distribution in most of the campaigns (Figure 3). Measurements at this distance allow the whole landfill to be considered as a single $CH_4$ emission source.

**Table 1:** Summary of all 21 MGL measurement campaigns and corresponding atmospheric conditions, averaged values of the meteorological and turbulence parameters over the campaign period (mean horizontal wind speed ($U$) and direction ($\theta$), the Obukhov length ($L$), surface friction velocity ($u_*$), and standard deviation of wind velocity fluctuations ($\sigma_u$, $\sigma_v$, $\sigma_w$), and Pasquill-Gifford-Turner (PGT) stability classes when the high-frequency measurements from the 3-D sonic were unavailable).

| No | Date | Primary (Sniffing) Gas Analyzer | Met Data Source | No of transects ABCD | No of transects EF | $U$ (ms$^{-1}$) | $\theta$ (°) | $L$ (m) | $u_*$ (ms$^{-1}$) | $\sigma_u$ (ms$^{-1}$) | $\sigma_v$ (ms$^{-1}$) | $\sigma_w$ (ms$^{-1}$) | PGT | Comments |
|---|---|---|---|---|---|---|---|---|---|---|---|---|---|---|
| 1 | 13-09-2016 | Picarro-G2203 | 2-D met | 6 | - | 5.52 | 145 | | | | | | D | |
| 2 | 17-11-2016 | Picarro-G2203 | 2-D met | 7 | - | 6.61 | 246 | | | | | | D | |
| 3 | 05-12-2016 | Picarro-G2203 | 2-D met | - | 14 | 2.0 | 96 | | | | | | C | westside EF roads |
| 4 | 11-08-2017 | Picarro-G2203 | 2-D met | 10 | 11 | 3.50 | 345 | | | | | | B | |
| 5 | 28-09-2017 | Picarro-G2203 | 2-D met | 10 | - | 2.17 | 179 | | | | | | B | |
| 6 | 06-10-2017 | Picarro-G2203 | 2-D met | 5 | 22 | 5.00 | 355 | | | | | | C | |
| 7 | 26-07-2018 | ABB-UGGA | Melun | 4 | 4 | 1.50 | 10 | | | | | | A | |
| 8 | 27-11-2018 | Picarro-G2203 | 3-D sonic | 9 | - | 1.85 | 335 | | 0.27 | 0.73 | 0.67 | 0.29 | | |
| 9 | 10-01-2019 | Picarro-G2203 | 3-D sonic | 22 | 12 | 3.53 | 2 | -1500 | 0.22 | 0.86 | 0.65 | 0.27 | | |
| 10 | 12-02-2019 | ABB-MGGA | Melun | 6 | 4 | 1.00 | 5 | | | | | | A | |
| 11 | 10-07-2019 | Picarro-G2203 | 3-D sonic | 4 | 2 | 2.65 | 25 | -9 | 0.29 | 1.21 | 1.54 | 0.39 | | |
| 12 | 02-08-2019 | Picarro-G2203 | 3-D sonic | 5 | 8 | 2.00 | 338 | -3 | 0.20 | 1.34 | 1.20 | 0.40 | | |
| 13 | 29-08-2019 | Picarro-G2203 (ABB-MGGA) | 3-D sonic | 5 | - | 2.59 | 303 | -17 | 0.29 | 1.04 | 1.12 | 0.35 | | Includes sniffing data |
| 14 | 13-09-2019 | Picarro-G2203 | 3-D sonic | 5 | 11 | 1.68 | 15 | -7 | 0.23 | 1.03 | 0.89 | 0.31 | | |
| 15 | 09-12-2019 | Picarro-G2203 | Melun | 8 | - | 9.00 | 345 | | | | | | D | |
| 16 | 05-02-2020 | Picarro-G2203 | 3-D sonic | 13 | 20 | 1.65 | 32 | -94 | 0.27 | 0.78 | 0.88 | 0.29 | | |
| 17 | 04-03-2020 | Picarro-G2203 (ABB-MGGA) | 3-D sonic | 18 | - | 4.96 | 1 | -205 | 0.43 | 1.10 | 1.11 | 0.47 | | Includes sniffing data |
| 18 | 04-09-2020 | Picarro-G2401 | 3-D sonic | 15 | - | 4.37 | 48 | -1 | 0.11 | 1.22 | 1.53 | 0.48 | | |
| 19 | 15-10-2020 | Picarro-G2401 | Melun | 9 | - | 4.00 | N | | | | | | C | |
| 20 | 01-12-2020 | Picarro-G2401 | 3-D sonic | 7 | 12 | 7.40 | 338 | -1297 | 0.53 | 1.46 | 1.15 | 0.71 | | |


| 21 | 08-12-2020 | Picarro-G2203 | 3-D sonic | 6 | 12 | 2.99 | 351 | -49 | 0.17 | 0.83 | 0.77 | 0.32 | | |


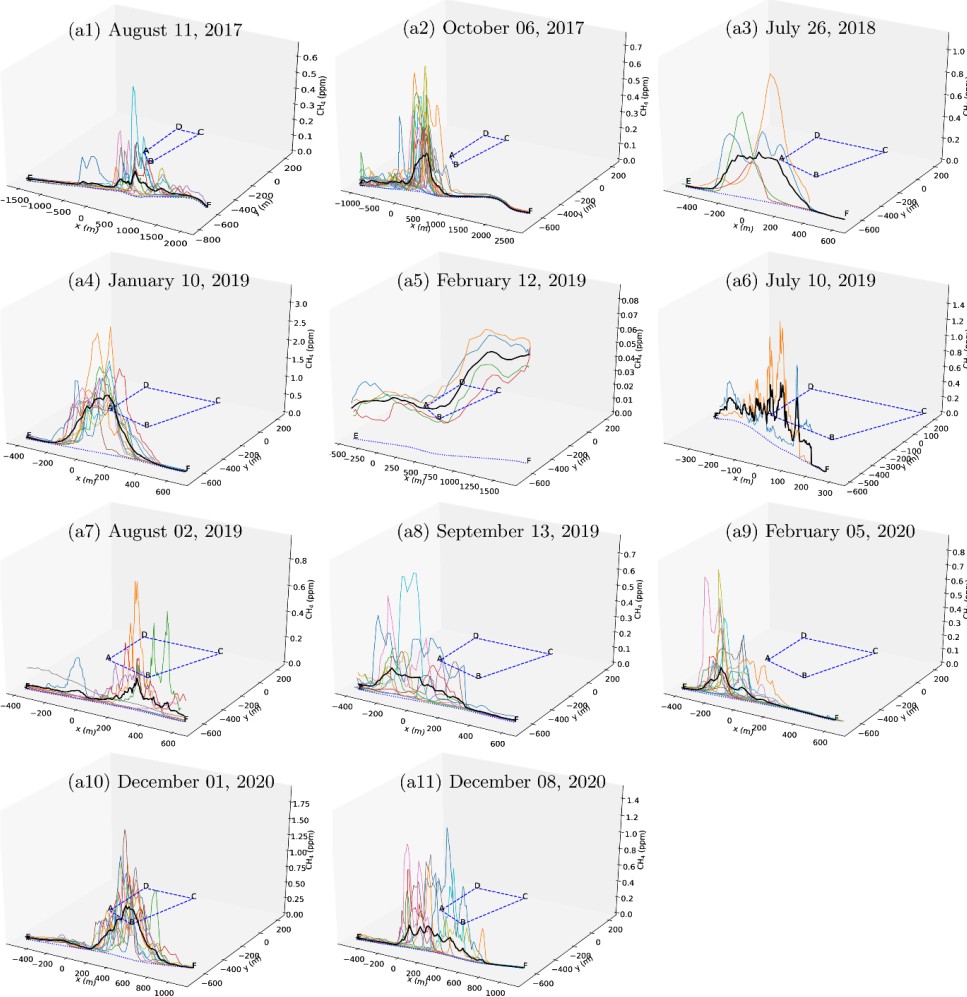

**Figure 3:** Enhancement of CH$_4$ mole fractions above the background in different plume cross-sections along the EF roads during different measurement campaigns. The black solid line shows the averaged CH$_4$ mole fractions computed from the different plume cross-sections in each campaign.

**3  Concentration mapping and leak detection: potential point and area sources within the landfill**

A rough knowledge "a priori" of (or assumptions on) the position and extent of the major CH$_4$ sources within the landfill are needed to set-up the inversion configurations or to strengthen the results from the inversions: whether the emissions correspond to a set of point sources or

relatively large area sources, and whether some areas tend to emit more than the others as a function of the period when the campaigns are conducted. Point and/or area sources within the



landfill originate from biogas pipes, well heads, damaged membranes, the biogas power plant, active and/or covered cells, leachate ponds, etc. However, a priori knowledge of the spatial distribution of methane emissions within the studied landfill was very limited before the sniffing campaigns. In a previous study to quantify emissions from the same landfill using mobile measurements from two campaigns (November 17, 2016 and December 5, 2016), Albergel et al. (2017) divided the site into five potential emission areas. This definition of potential area sources was based on rough information from the landfill operators and did not account for potential emissions from the biogas valorization plant, leachate ponds, or from the pipe/well network. In this study, we conducted two sniffing campaigns by foot and relied on these to identify the principal methane hotspots (Section 3.1). Furthermore, we performed a detailed analysis of the measurements conducted at the borders of the landfill along the ABC road from different mobile campaigns combined with corresponding wind speeds and directions to explore if these could provide some insights about the potential $CH_4$ emission sources within the landfill, or on their spatial representativity.

### 3.1 Identification of $CH_4$ hotspots from the sniffing campaigns

We analyzed measurements from the foot-based sniffing campaigns on August 29, 2019 and March 04, 2020 to identify the potential methane hotspots and their source origin. It is important to recognize that during these sniffing campaigns, $CH_4$ mole fraction measurements were often obtained very close to the source, and therefore, high mole fraction observations do not necessarily correspond to equally large fluxes.

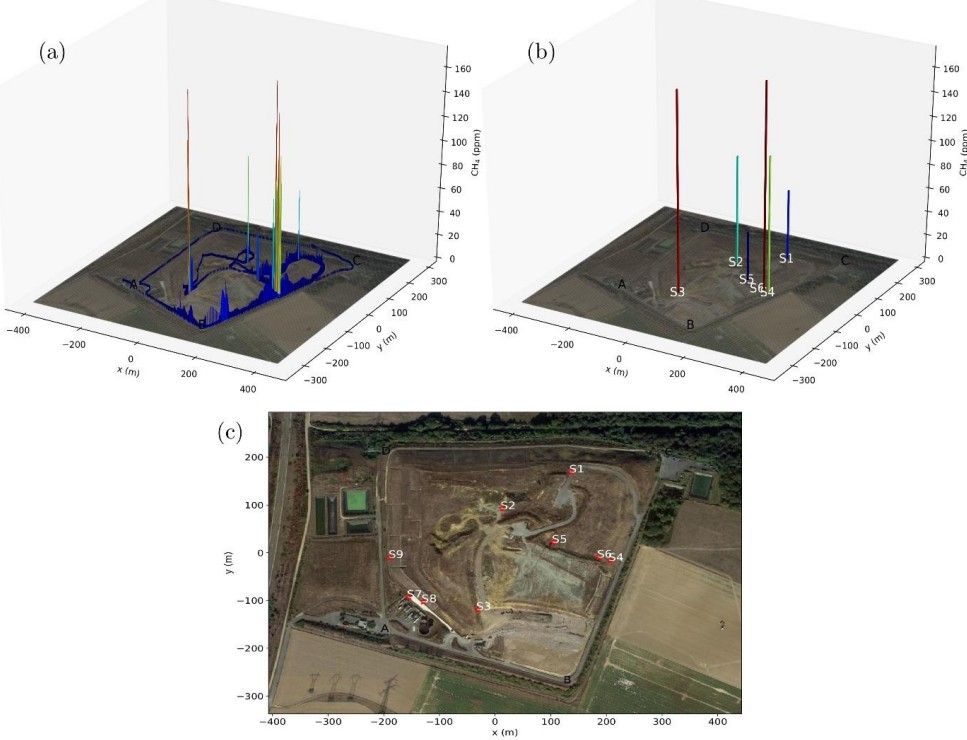

**Figure 4:** (a) Observed $CH_4$ mole fractions from sniffing on August 29, 2019 within the studied landfill using an ABB MGGA along with a GPS module, and (b) high $CH_4$ mole fraction peaks from the sniffing data are assumed to correspond to the main emission hotspots: six of them are identified during this campaign (S1 to S6). Figure

(c) shows the locations of a total of 9 emission hotspots (S1 to S9) identified from both the sniffing campaigns on August 29, 2019 and March 04, 2020. The underlying aerial photograph background images are taken from Google Earth (© Google Earth).

Figure 4(a) shows the spatial distribution of $CH_4$ mole fractions along the measurement path from the sniffing campaign on August 29, 2019. Six locations with high $CH_4$ peaks, at least ~30 m apart from each other, were identified (Figure 4(b)). These locations were examined with a detailed map of the biogas collection pipes, wells, leachate ponds, gas processing facility, etc. to identify their source origin. Based on this analysis, we found that the two hotspots S1 and S2 are near biogas network purges, S4 is located near a biogas network well, S5 is at the location of a bioreactor tank, and S6 is near a leachate bioreactor/biogas purge where landfill gas is removed from the landfill cells and also close to a major junction of biogas pipes. The hotspot S3 is near to a leachate well and also downwind of a biogas network and a well. The methane peaks in different mobile plume transects on the ABC road during this campaign (Figure S1.14(c)) are consistent with these six hotspots within the landfill.

The results of the sniffing campaign on March 4, 2020 confirmed $CH_4$ hotspots at similar locations (S1 to S6) to those observed on August 29, 2019. Additional measurements obtained near biogas network wells, a biogas network purge (S9), and two along a drainage gutter behind the biogas power plant (S7 and S8) (Figure 4(c)), indicated three more hotspots with measured methane mole fractions in the range of 60 to 800 ppm. Therefore, we identified a total of 9 hotspots (S1 to S9) from the analysis of the two sniffing campaigns (Figure 4(c)). These 9 potential methane emission point sources were used in the inversion tests to estimate their emissions. It is important to note that the rapid ability to identify leaks from these sniffing campaigns provides an opportunity for site operators to easily diagnose methane emissions and take actions to reduce them, whilst also increasing the yield of $CH_4$ that is captured and available for sale or use on-site.

### 3.2 Directional information on potential $CH_4$ emission sources from the plume cross-sections along the ABC road

Other than the two "sniffing" campaigns, which offered a snapshot insight into localized emission sources, little information is available about the presence and characteristics of emissions within the landfill. To gain more insights, we analyzed the plumes collected along the ABC road under various wind conditions from different campaigns. A first analysis of the ABC measurements from different mobile campaigns indicates a similarity of the plume cross-sections along this road despite changes in wind direction from one campaign to the next (Figures S1.1-S1.21). It indicates that these measurements are more representative of both the localized leakages from pipes and wells near to these roads than of the emissions from the greater landfill. Furthermore, we constructed bivariate polar plots from all the plume cross-sections along the ABC road from the 11 campaigns between July 2018 and December 2020 where on-site wind data from the 3-D sonic anemometer was available (Table 1; Figure 5). These bivariate polar plots can provide useful directional information on the potential emission sources and may help to identify the presence and characteristics of these sources (Carslaw and Beevers, 2013).

The bivariate polar plots from the ABC plume cross-sections are constructed in the following way. The ABC road is divided into seven segments (Seg-1 to Seg-7) and for each segment, $CH_4$ mole fraction enhancements above the background are averaged over the duration of each transect in that segment. Wind speed and direction measurements are averaged over durations starting from one minute prior to a transect in a segment until the end of the transect in that segment. The averaged wind speeds, wind directions, and mole fraction data are partitioned into wind speed and direction bins and the mean $CH_4$ mole fractions are calculated for each



bin. The mean $CH_4$ mole fractions in each wind speed-direction bin are plotted using polar
coordinates. We used wind direction intervals at 22.5° and wind speed intervals at 1 ms$^{-1}$ for
binning the data in each bivariate polar plot. The mean $CH_4$ mole fractions, calculated in wind
speed-direction bins with limited data points, such that those with 1, 2, and 3 points, are down-
weighted with the weights 0.25, 0.50, and 0.75, respectively (Carslaw and Beevers, 2013).

Figure 5 shows the seven bivariate polar plots in each segment (Seg-1 to Seg-7) along the ABC
road. These bivariate plots provide different directional information on the likely methane
emission sources contributing to the methane mole fractions in different segments. The polar
plot in Seg-1 suggests that at least two small sources were present just north of this segment
(near the biogas power plant), as indicated by the elevated mean $CH_4$ mole fractions in the bins
with northerly winds when wind speeds were moderate (~4-6 ms$^{-1}$). One of the "sniffing"
campaigns on March 04, 2020 also identified two hotspots (S7 and S8) in this area along a
drainage gutter behind the plant (Figure 4(c), Section 3.1). The polar plots in Seg-2 to Seg-6
indicate multiple emission sources within the whole landfill with potentially high emitting
sources corresponding to Seg-2 and Seg-3 and small sources corresponding to Seg-4 to Seg-6.
The high $CH_4$ mole fractions in the northerly wind directions bins in Seg-2, 3, and 4 are strongly
influenced, and most probably caused by the last, uncovered, active cell of the landfill in the
south-east corner. The plots in Seg-2 to Seg-6 also indicate some local emission sources near
the roads, from the high mean $CH_4$ mole fractions in the bins of low wind speeds. High $CH_4$
mole fractions in the northerly wind directions bins in the polar plot in Seg-5 indicate potential
$CH_4$ emission sources that could correspond to hotspots S4 and S6, identified by the sniffing
campaigns. The polar plot in Seg-7 has a small number of data points and does not indicate any
major source upwind of the segment. The polar plots in Seg-4 to Seg-6 also show some
unexpected elevated mean $CH_4$ mole fractions in the bins with north-east wind directions and
moderate wind speeds, which indicates potential emitting sources in the north-east, outside the
landfill. However, there are only agriculture farms in the north-east of these segments of the
landfill, where we do not expect any major methane sources, except only minor methane
emissions from using fertilizers or manure, which are unlikely to explain such enhanced $CH_4$
mole fractions. The ABC road follows the border of the landfill with localized leakages from
pipes and wells near to these roads and half of the road segments (A-B and B-C) are adjacent
to a steep ridge in the south-east. Therefore, recirculation of the wind flow due to these ridges
and the complex landfill topography may explain these observations. The transport of a plume
in a complex flow field along the B-C road, especially when the wind is from the north-east to
south-east directions does not follow the observed mean wind directions. As the air from north-
easterly or easterly wind directions is deflected against the ridges of the landfill, it is possible
that high $CH_4$ mole fractions may be measured along the B-C road, even though the air would
appear to originate from outside the landfill.

This analysis of the polar bivariate plots substantiates the evidence of methane hotspots
identified from the sniffing campaigns (Section 3.1). Furthermore, these results question the
ability of the ABC measurements, which might be strongly impacted by sources located along
the roads, to spatially represent the emissions from the greater landfill. This would hamper the
use of this data for inverting landfill emissions. The complex atmospheric transport along the
ridge also raises large uncertainties in inversions using this data with a simple Gaussian model
(Section 5.1).

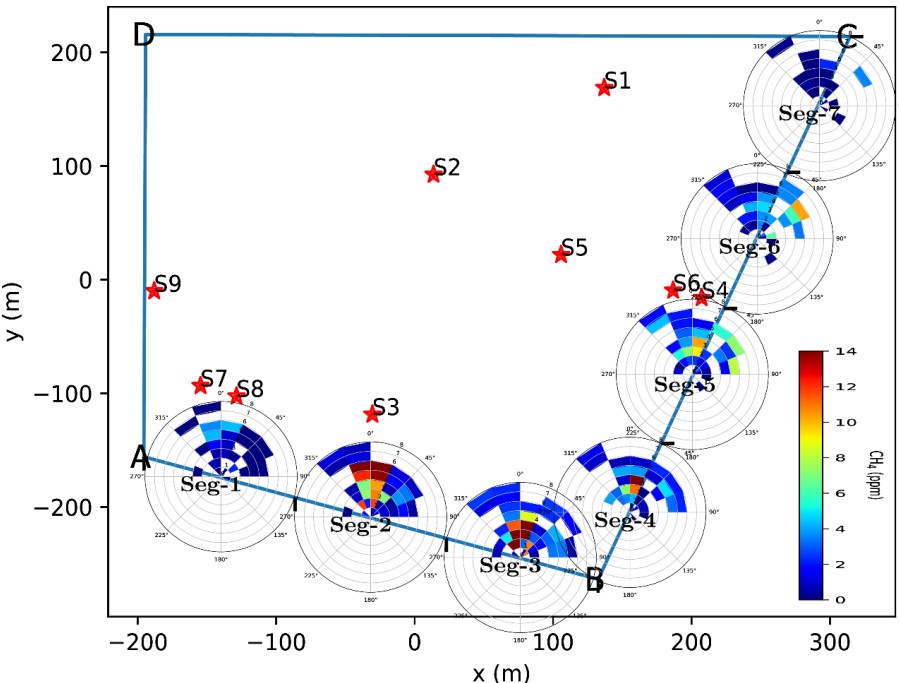

**Figure 5:** Bivariate polar plots of the mean $CH_4$ mole fractions enhancement above background in seven
equidistant segments (Seg-1 to Seg-7) obtained from mobile transects during 11 campaigns between July 2018
and December 2020 along the ABC road. Each polar plot in a segment uses the values of $CH_4$ mole fractions
averaged over the duration of the part of each mobile transect in that segment. Nine red stars (S1 to S9) indicate
the key $CH_4$ hotspots identified from two sniffing campaigns.

### 3.3  Definition of potential emission sources within the landfill for inversion tests

The detection of hotspots during the two sniffing campaigns within the landfill (Section 3.1)
and the analysis of the mobile measurements along the ABC road in different wind conditions
from different campaigns (Section 3.2) indicate that landfill methane emissions come from a
combination of area and point sources. Consequently, we develop several inversion
configurations, one of which defines the potential sources as 9 hotspots identified from the
sniffing (Section 3.1, Figure 4(c)), while others correspond to the area sources (Figure 6). The
analysis of the $CH_4$ enhancements measured along the ABC road provided only qualitative
directional information on the area and/or point sources within the landfill (Section 3.2).
However, due to the complex nature of the landfill and the spatiotemporal variability of
emissions, it is uncertain whether we have detected all the hotspots through sniffing, and
identifying the area sources of emissions with more dispersed emissions is exceedingly
challenging. As a consequence, we have chosen to define a set of large area sources with
uniformly distributed methane emissions for inversions. Thus, we defined 6 potential emission
source regions, i.e. 6 area sources that include the biogas power plant (A-1) and the five cells
(A-2 to A-6) within the landfill (Figure 6).



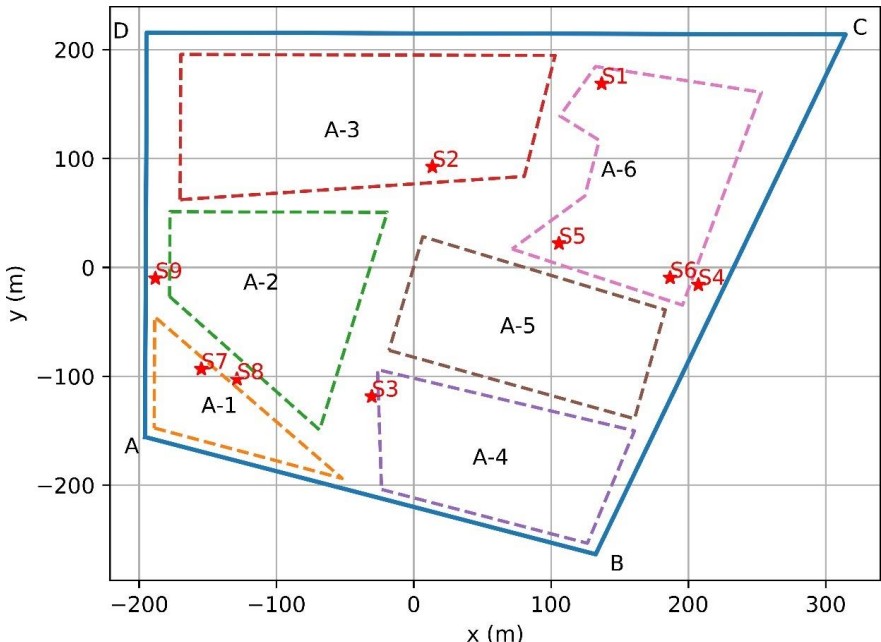

**Figure 6:** The six potential area sources (boxes, A-$i$, $i$=1,..,6) in the configuration of the inversion, defined as the biogas valorization plant (A-1) and the five cells (A-2 to A-6). Nine red stars (S1 to S9) indicate the $CH_4$ hotspots identified from two sniffing campaigns.

## 4    Atmospheric inversion of landfill methane emissions

We used a simple atmospheric inversion framework to quantify $CH_4$ emissions from multiple potential sources within the landfill using the MGL measurements. The inversion exploits some of the basic theoretical and practical components of the approaches described in Kumar et al. (2022, 2021) and Ars et al. (2017) and uses the assumption about the characterization of potential $CH_4$ emissions sources from Section 3. We used a Gaussian plume dispersion model

designed for single point sources from Kumar et al. (2021, 2022) for estimating emissions from the 9 $CH_4$ hotspots as point sources (Section 3.2) and adapted the same Gaussian model to simulate the dispersion from area sources when estimating emissions from the 6 area sources (section 3.3). Details on the Gaussian plume model equations for a point source dispersion and their adaptation to an area source dispersion are provided in the supporting information

(Section S2.1). We describe two different approaches to formulate the Gaussian model for an area source dispersion:  *method*-1: a very simple approach by modifying the lateral plume spread to the total plume width as a sum of the plume spread due to atmospheric turbulence and of the additional initial spread due to the source size (Section S2.1.1(a)), and *method*-2: by decomposing an area source into multiple point sources and superimposing the modelled

Gaussian plumes from all of these point sources to compute the average plume from that area (Section S2.1.1(b)).

When on-site measurements from a meteorological station (3-D sonic anemometer or 2-D) were available, the Gaussian model was driven by the averaged wind direction given by the meteorological data. When relying on the data from the Melun station, the mean wind direction

was approximately taken as a direction from the center of the landfill to the location of the maximum averaged $CH_4$ mole fraction (Kumar et al., 2021). This wind direction approximation was deemed more representative of the landfill rather than the Melun wind direction and we



evaluated the effect of this approximation on the estimates in Section 5.2. In all cases, the model is driven by the effective mean wind speed from the meteorological data (section S2.1).

The dispersion parameters in the Gaussian model are defined by the standard deviations of the velocity fluctuations ($\sigma_v$, $\sigma_w$) when we have the high-frequency 3-D sonic data available and in other cases (four campaigns), based on the Briggs dispersion formulas for flat terrain (Briggs, 1973) corresponding to the defined PGT stability classes. We used the Gaussian model to simulate the plume (called response function) of each potential $CH_4$ source separately at the

measurement locations with atmospheric conditions observed during the averaging periods of ABC and/or EF plume transects and using a unitary emission rate (1 kg/s). A response function defines a linear relationship between the emission rate of a potential source and the concentration at a measurement location.

We used a non-negative least-squares minimization approach to formulate the inverse problem

for the quantification of unknown emissions of multiple potential emission sources. The details of this inversion procedure are provided in the supporting information (Section S2.2). The principle of the inversion process is to minimize the root sum squared misfits between the averaged observed and modelled mole fraction enhancements in the plumes from the multiple potential sources. These inversions rely on a priori information about the potential emission

sources (e.g., number, type, location, size, and/or shape), the response functions simulated with the Gaussian model for each potential emission source, and the observation vectors of the measured and modelled plumes. We employed two options to define the observation vectors in the inversion. The first observation vector ($\mathbf{\mu_{pt}}$) is defined as the averaged $CH_4$ mole fractions at the measurement locations along the roads. Since we have to estimate multiple sources of

methane emissions within the landfill site, following Ars et al. (2017), we discretize the roads into multiple segments of equal length and for each segment, the integrated areas under the averaged $CH_4$ mole fractions are used to define a different observation vector ($\mathbf{\mu_{SI}}$). This approach reduces the tendency of the inversion to over-fit turbulent patterns within the plume. We divide the plumes into a different number of segments on the ABC and EF roads with 50

m and 100 m distance intervals, respectively. More information about these observation vectors is given in supporting information (Section S2.2).

For both ABC and EF roads, we conducted six inversion tests using two types of observation vectors ($\mathbf{\mu_{pt}}$ and $\mathbf{\mu_{SI}}$) for three source configurations. The source configurations involve 9 point sources (hotspots identified from the sniffing campaigns, as discussed in Section 3), and 6 area

sources (Section 3) modelled by two different area source adaptations of the Gaussian model (*method*-1 and *method*-2).

## 5 Results

We conducted inversion tests for all of the selected campaigns when the wind conditions allowed us to obtain plume cross-sections on ABC (near field) and/or EF roads (far field).

However, it is challenging to model the plume cross-sections along the ABC road using a simple Gaussian plume dispersion model and, therefore, to invert the site emissions based on the data measured on this road. The dispersion of $CH_4$ from the potential sources to the ABC road is highly sensitive to the complex topography of the landfill, which is not taken into account in the Gaussian modeling. The vicinity between this road and the potential sources in

the landfill makes these measurements also highly sensitive to factors such as the a priori information on the location and extent of the potential emission sources, while Section 3 shows that we can hardly provide a precise distribution of the sources within the landfill. Finally, Section 3.2 highlighted our lack of understanding of the spatial representativity of the measurements along the ABC road. The inversions using data from the ABC road are thus

likely hampered by large uncertainties and need to be analyzed cautiously, but they may





provide insights into the spatial distribution of the emissions. On the contrary, the shape of the observed averaged plume along the EF roads is almost unimodal in most of the campaigns and the Gaussian model should be more suitable for the modelling of the transport over the distance between the potential sources within the landfill and the EF road. Therefore, we do not expect
the inversions based on the data from the EF road to provide better insights on the spatial distribution of the emissions compared to the ABC road; however, we expect them to provide much more robust estimates of the total emissions from the landfill than those based on the data from the ABC road.

The campaign of January 10, 2019 is taken as an example to illustrate the analysis of the data
and the inversions. The Gaussian model for this campaign is driven by the measured meteorological and turbulence parameters from the on-site 3-D sonic anemometer data. Wind directions during this campaign were mainly from the north which allowed us to get 22 and 12 $CH_4$ plume cross-sections on the ABC and EF roads, respectively (Table 1, Figure S1.10). Furthermore, the absolute magnitude of the Obukhov length ($L$) computed from the 3-D sonic
data is greater than 1000 m (Table 1) which suggests neutral atmospheric stability conditions during this campaign. The averaged $CH_4$ mole fraction plume along the ABC road shows multiple peaks (Figure S1.10); whereas, the averaged plume along the EF road is unimodal (Figure 3(a4)). Observed enhancements of the averaged $CH_4$ plumes above the background, reached up to ~25 ppm and ~1.5 ppm, along the ABC and EF roads, respectively.

The division of the observed and modelled plumes over sub-segments of ABC and EF roads (to build $\mu_{SI}$) from the January 10, 2019 campaign is illustrated in Figures 7(a1) and (b1) and Figure 8 (a1) and (b1), respectively. Figures 7(b1) and 8(b1) illustrate a comparison of the modelled plumes with *method*-1 and *method*-2 from each potential area source at the measurement roads ABC and EF, respectively. For the ABC road, the shapes of modelled
plumes from two different methods for the area sources A-1 to A-3 (which are a little farther from the ABC road) are approximately similar. However, noticeable differences in the shapes and magnitudes (i.e. horizontal spread) can be seen in the modelled plumes from the sources A-4 to A-6, which are closer to the measurement road ABC. The *method*-2 plumes are slightly narrower and have a larger maximum than those from *method*-1. Figure 8(b1) for the EF
measurements shows that the behavior of modelled plumes from both methods is approximately similar and unimodal. Some differences can be noticed in terms of magnitude and width, with *method*-2, plumes being slightly narrower and having a larger maximum than *method*-1 (Figure 8(b1)).

### 5.1 Emission estimates using ABC road measurements

Figure 7 illustrates the inverted emissions using measurements from the ABC road for the campaign of January 10, 2019. The total estimated $CH_4$ emissions using $\mu_{pt}$ and $\mu_{SI}$ in the inversion tests with 9 hotspots are 22.94 and 22.82 t $CH_4$/d, respectively. The total emissions using $\mu_{pt}$ (and $\mu_{SI}$) and using 6 area sources with *method*-1 and *method*-2 are 12.98 (13.09) t $CH_4$/d, and 13.83 (13.56) t $CH_4$/d, respectively. Figures 7(a2)&(b2) show that the fit between
the observed and modeled $\mu_{pt}$ from the Gaussian model using the corresponding emission estimates with 6 areas sources is slightly better than that from the 9 hotspots. The inversion using 9 hotspots assigns the estimated emissions to the three point sources that lie in two source areas A-6 and A-4 only (Figure 7(a3)). Whereas, the estimated emissions from the inversion using 6 area sources are approximately equally distributed to three area sources A-4, A-5, and
A-6 (Figure 7(b3)). It is noticed that the total estimates are weakly sensitive to the observation vector $\mu_{pt}$ or $\mu_{SI}$. However, the discrepancy between the estimated emissions obtained with different definitions of the potential emission sources, and also from different implementations of area sources (*method*-1 and *method*-2) in the inversion tests is noticeable. The absolute



differences between the estimated emissions using 9 point sources and 6 area sources in the
inversions are ~10 and ~9 t CH$_4$/d for *method*-1 and *method*-2, respectively.

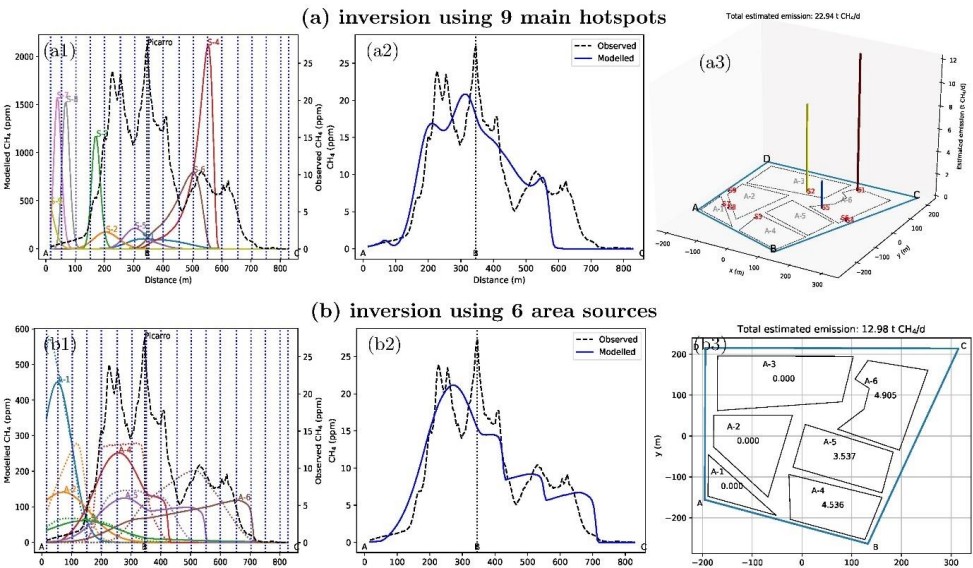

**Figure 7:** An example of modelling the individual plumes and emission rates from the inversion tests using (a) 9
main hotspots and (b) 6 area sources with $\mu_{pt}$ from the measurements obtained along the ABC road on January
10, 2019. From left to right in each row, first to third columns plots respectively show (1) the average CH$_4$ mole
fraction enhancements above the background (black dashed line, right Y-axis) and modelled response functions
(solid colored lines for *method*-1 and the same colored dotted lines for *method*-2, left Y-axis) for each potential
source, (2) the fit between the observed (black dashed lines) and modelled (blue sold lines) CH$_4$ mole fraction
enhancements, and (3) estimates of the CH$_4$ emissions (t CH$_4$/d) for each of the potential sources. Vertical black
dotted lines in the first column figures show the point of division of the roads into sub-segment over which the
averaged mole fractions are integrated to define $\mu_{SI}$.

For most of the selected campaigns using data from the ABC road, we observed a similar
behavior of the estimated CH$_4$ emissions from different inversion tests as from the results from
the January 10, 2019 campaign. The estimated emissions using ABC data from different
campaigns vary between ~2 to ~36 t CH$_4$/d using 6 area sources and ~4 to ~23 t CH$_4$/d using 9
point sources in different inversion tests (Figure S2.1). The estimates show large biases in the
order of magnitudes between total methane emission estimates from different tests. The large
differences in the inverted total CH$_4$ emissions using different definitions of the potential
sources in the inversion tests show a high sensitivity of the estimates to a priori information
about potential sources.

We analyzed the spatial distribution of methane emissions estimated from the inversions using
ABC measurements. Figure S2.16 shows the spatial distributions of the estimated CH$_4$
emissions attributed to the individual source regions from the inversions using six area sources
and $\mu_{pt}$ from the ABC measurements from all the selected campaigns. This shows that the two
source areas A-1 and A-2 have negligible contributions to the total estimated methane
emissions. Emissions from sources A-3 to A-6 are more regularly inferred from most of the
campaigns. Emissions from A-3 are variable and may indicate a highly variable source, while
emissions from A-4 are more consistent, which may be expected as this area of the landfill was
active during this time. High methane emissions attributed to the A-6 source region during

some of the campaigns may be emitted from the methane hotspots identified from the foot sniffing campaigns near the biogas network purges, biogas network well, and bioreactor tank (Section 3.1).

### 5.2 Emission estimates using EF road measurements

For the campaign of January 10, 2019, the total estimated $CH_4$ emissions using $\mathbf{\mu_{pt}}$ and $\mathbf{\mu_{SI}}$ in the inversions with 9 hotspots are 4.50 and 3.98 t $CH_4$/d, respectively. The total estimated $CH_4$
emissions using $\mathbf{\mu_{pt}}$ (and $\mathbf{\mu_{SI}}$) and 6 area sources with *method*-1 and *method*-2 are 4.44 (4.41) t $CH_4$/d, and 4.16 (4.18) t $CH_4$/d, respectively. Figures 8(a2)&(b2), respectively, for 9 hotspots and 6 area sources with $\mathbf{\mu_{pt}}$ in the inversions, show a good agreement between the observed and modelled $CH_4$ mole fractions from the dispersion model using the corresponding inverted emissions. The estimated $CH_4$ emissions from the inversions with *method*-1 and *method*-2 for
an area source implementation in the Gaussian model have a small percent difference of ~6% using either $\mathbf{\mu_{pt}}$ or $\mathbf{\mu_{SI}}$. The inversion results using EF measurements are weakly sensitive to the defined observation vectors $\mathbf{\mu_{pt}}$ and $\mathbf{\mu_{SI}}$ with ~12% and less than ~1% percent differences in flux estimates from 9 hotspots and 6 area sources, respectively. The total estimated methane emissions using 9 hotspots and 6 area sources with $\mathbf{\mu_{pt}}$ had small percent differences of ~1%
and ~8% for *method*-1 and *method*-2, respectively. Figure 8(a3) shows that in the inversion using 9 hotspots, the estimated emissions are distributed only to three point sources in two source areas, A-6 and A-4. In contrast, the inversion using 6 area sources assigns the estimated emissions primarily to A-6, with small contributions from A-5 and A-4, as shown in Figure 8(b3).

We conducted another sensitivity analysis of the inversion results with respect to a different definition of the five rectangular potential area sources defined within the five cells (Figure S2.2), proposed by Albergel et al. (2017). Using these five area sources and with $\mathbf{\mu_{pt}}$ obtained from the EF measurements from January 10, 2019, the total estimated emissions (4.24 t $CH_4$/d and 4.19 t $CH_4$/d with *method*-1 and *method*-2, respectively) (Figure S2.3) have small percent
differences (~4% and ~1%) from the total estimated emissions (4.44 t $CH_4$/d and 4.16 t $CH_4$/d for *method*-1 and *method*-2, respectively) obtained using 6 area sources in inversions. In order to analyze the effect of the approximated wind direction (Section 4) on inversion results when relying on the meteorological data from Melun met station in the Gaussian model, we tested this assumption for the campaign on January 10, 2019, where instead of using actual observed
wind direction, we forced the model to use the wind direction approximation. With $\mathbf{\mu_{pt}}$, total estimated emissions of 4.03 t $CH_4$/d and 3.80 t $CH_4$/d using 9 hotspots and 6 area sources, respectively, have ~11% and ~15% percent differences to those obtained using the actual observed mean wind direction from the local 3-D sonic anemometer (4.50 and 4.44 t $CH_4$/d, respectively). Overall, different sensitivity tests using EF measurements from January 10, 2019
indicate that the percent differences between the total estimated emissions range from less than 1% to ~15%. This suggests that the total estimated emissions exhibit weak sensitivity to different input parameters in the inversion tests.

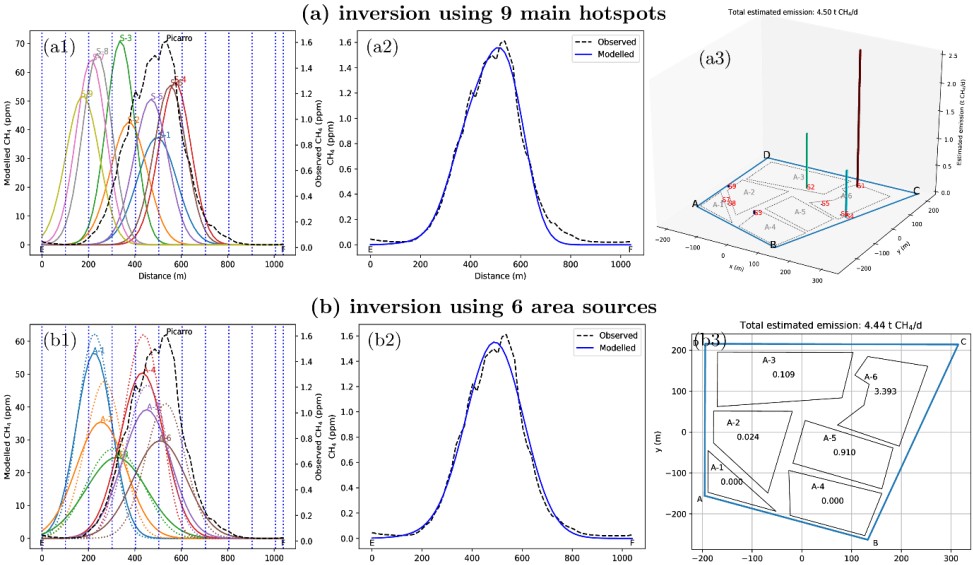

**Figure 8:** Same as for Figure 7, but for the measurements obtained along the EF road on January 10, 2019.

Figure 9 shows the estimated total methane emissions from the studied landfill using $\mu_{pt}$ obtained from the EF measurements, from the 11 campaigns where sampling was conducted on the southern EF road. These estimations are based on using 9 hotspots as prior point sources and 6 potential area sources, with two different methods (*method*-1 and *method*-2) for area source implementation in the Gaussian model. Figures S2.3-S2.14 in SI-2 present more details

about these inversion results. The total $CH_4$ emissions using 9 hotspots from the inversions vary from 0.44 t $CH_4$/d (February 05, 2020) to a maximum of 6.90 t $CH_4$/d (December 01, 2020), with an average emission of 2.24 t $CH_4$/d. These estimates are similar to the estimated emissions obtained using 6 area sources with *method*-1 (and *method*-2) which vary from 0.34 (0.34) t $CH_4$/d to 7.04 (6.30) t $CH_4$/d, with an average value of 2.07 (2.00) t $CH_4$/d.

Similar to the inversion results using EF measurements from the January 10, 2019 campaign, the results from different inversion tests using three different definitions of the potential emissions sources, two observation vectors, two different implementations of the area sources in the Gaussian plume model show that the percent differences between the total estimated emissions from different combinations of these tests averaged over the 11 campaigns ranged

from ~1% to ~15%. This analysis shows that the emission estimates using EF measurements are weakly sensitive to the different definitions of potential emission sources, observation vectors, and other parameters considered in the inversion tests. Thus, based on this analysis, we consider that the total estimated methane emissions using the EF measurements are robust. The estimates obtained through ABC measurements in inversions are much higher compared

to the estimates from EF road. These ABC estimates are highly sensitive to different characterizations of potential sources (Section 5.1), due to various factors such as the complex landfill topography, inability to account for it in the Gaussian model, our limited understanding of the spatial representativity of potential sources, short distances between measurements and potential sources, etc. This weakens our confidence in the estimates derived from the data

collected along ABC road. Therefore, we rely on the estimates obtained using measurements from the EF road for the estimation of landfill methane emissions.

We also analyzed the spatial distribution of the estimated emissions from the EF road measurements. Figure S2.17 shows that the inversions using EF measurements assign a significant proportion of net methane emissions to the A-6 area source (Figure 6), along with 725 some contributions from A-4, and A-5. EF measurements additionally attributed a small part of total methane emissions to the A-1 source area which includes the biogas plant and was not detected by the inversions using ABC measurements.

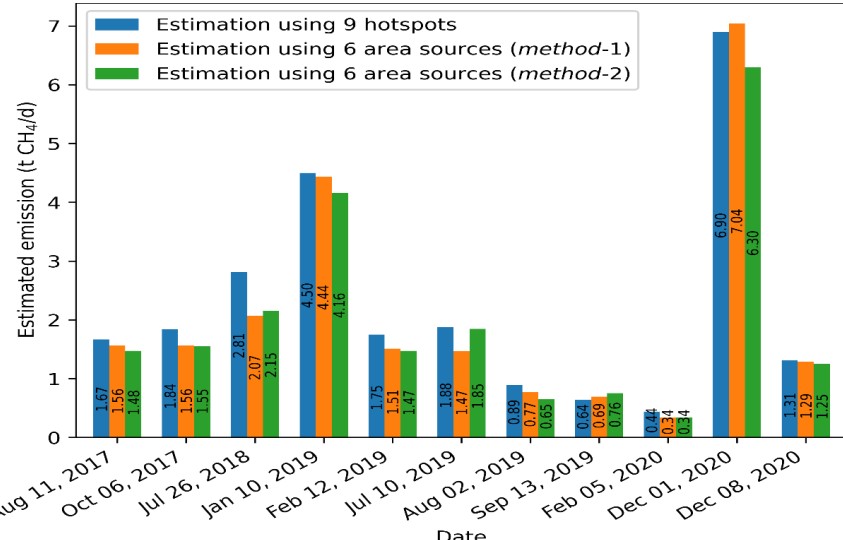

**Figure 9:** Summary of the total estimated $CH_4$ emissions using the observation vector $\mu_{pt}$ obtained from EF road 730 data and using 9 hotspots from sniffing as point sources and 6 area sources with two different methods (*method*-1 and *method*-2) for area source implementation in the Gaussian model.

## 6    Discussion

. The averages of total $CH_4$ emissions using data from EF measurements from all 11 campaigns (where suitable MGL sampling was conducted) vary from ~2.0 t $CH_4$/d to ~2.2 t $CH_4$/d in 735 different inversion sensitivity tests. It indicate that the use of remote mobile plume cross-section measurements are suitable for quantification of the total methane emissions from the site, which are weakly sensitive to the characterization of the potential emission sources and other influencing parameters like the observation vectors, wind directions, etc. On the other hand, the inversion tests performed with sampling from the landfill perimeter (ABC), show a 740 high sensitivity of the estimates to the spatial distribution of the potential emission sources and other parameters in the inversions. It highlights the difficulties in exploiting ABC measurements to estimate $CH_4$ emissions using a simple Gaussian plume model due to the model's inability to consider complex landfill topography and lack of precise information about potential emission sources. Thus, estimates using the ABC road were deemed poorly 745 representative of actual landfill methane emissions.

The total $CH_4$ emissions using data from the EF road show a large temporal variability (~0.4 t $CH_4$/d to ~7 t $CH_4$/d) in landfill methane emissions (Figure 9). The emission sources and thus the methane emissions from an active landfill can vary greatly even over a small period of a few days. For example, total methane emissions on December 08, 2020 (~1.25 t $CH_4$/d) were 750 far smaller than on December 01, 2020 (~7 t $CH_4$/d), despite a one-week interval between these two sampling campaigns and despite the fact that measurements were conducted during



daytime hours of between 11:30 to 12:30 UTC in both campaigns. Thus, more mobile campaigns on the EF road are required to more accurately monitor and to better understand temporal variabilities of landfill methane emissions. Note, that the estimates from each of the

selected campaigns are based on the measurements spanning an order of one to two daytime hours. However, different atmospheric conditions and landfill activities during nighttime and other daytime hours may contribute to a diurnal pattern in landfill emissions (Sonderfeld et al., 2017). To better understand the diurnal variability of landfill methane emissions, we need to monitor the emissions at a higher temporal resolution. For this, continuous automated

measurements at a certain distance of the site over long periods are required, which is impractical using a labor-intensive MGL, unless the MGL were permanently installed in a vehicle that travelled along the EF transect frequently. Continuous $CH_4$ mole fractions measurements from a network of fixed sensors around a site alongside meteorological measurements can provide an alternative, to develop an automated monitoring system to

monitor long-term landfill methane emissions at higher temporal resolution. However, the deployment of a dense network of high-precision sensors is still limited by cost. Recently, Riddick et al. (2018) utilized a single-point continuous $CH_4$ measurement that was sampled ~700 m downwind from a landfill, and they combined this with a Lagrangian particle model to estimate the methane emissions at a high temporal resolution. A similar approach can be

applied to monitor landfill methane emissions for short and long-term temporal variability studies. Such an approach could be complemented by other techniques, such as MGLs, which may provide complementary information on the spatial variability of sources within the landfill, and which may be more suited to leak detection and mitigation.

A limitation of our inversion approach is that it does not diagnose explicit estimates of the

uncertainties in the estimated $CH_4$ emissions. Extrapolating the results obtained with a similar approach applied to controlled $CH_4$ release experiments during TADI-2018, and TADI-2019 campaigns (Kumar et al., 2022, 2021), we assume that our emission estimates from the EF road have a level of uncertainty of ~30% . The errors diagnosed during TADI's controlled release experiments were mainly applicable for flat terrain conditions. Here, much of the plume

dispersion from the landfill to the measurements transects occurs over flat terrain. However, the landfill itself correspond to a complex topography. We currently lack information on the errors from Gaussian plume dispersion models when applied to such a terrain, making it difficult to provide a more robust diagnostic of uncertainties in our estimates.

Several studies have shown that the temporal variability of landfill methane emissions is driven

by absolute or temporal gradients of some meteorological parameters, especially atmospheric pressure (Aghdam et al., 2019; Czepiel et al., 2003; Kissas et al., 2022; Poulsen Tjalfe G. et al., 2003; Xu et al., 2014). A limited number of studies, like Riddick et al. (2018), have demonstrated a very weak negative or no clear relationship between landfill $CH_4$ emissions and changes in atmospheric pressure. We also analyzed this emission-pressure or emission-

temperature relationships using the estimated $CH_4$ emissions from the EF road measurements and the atmospheric pressure and temperature measured at Melun station (Figure S2.15). We observed a weak negative correlation of landfill methane emissions with atmospheric pressure (R = -0.10) and a slightly stronger negative correlation with atmospheric temperature (R = -0.30) (Figure S2.15). Riddick et al. (2018) discussed several possible contributing factors to

this weak emission-pressure relationship, such as on-going landfill operations on an active landfill during a measurement campaign and emission data gaps. These are reasonable contributory factors in our case of the studied active landfill, as the sample size of our landfill emission estimates is very small with large data gaps between emission estimates.



For near-landfill measurements on the ABCD road, methane plumes coming from the sources
within the landfill are generally not well mixed either horizontally or vertically as they are too
close to the emission sources. The discrete landfill emission sources at higher elevations may
not be detected within these measurements, with the sampling air intake at ~2 m above the
ground surface. Recirculation of the wind flow due to complex landfill topography affects the
transport and dispersion of mixing methane plumes at the measurement positions, which is
difficult to simulate with a simple Gaussian plume model that considers spatially homogeneous
flow. Thus, the estimation of methane emissions using these measurements requires a more
complex model that can resolve the flow-field and turbulence induced by the complex
topography of the landfill. Computational fluid dynamics (CFD) models are more suitable for
such applications, which have been used to simulate high-resolution flow-fields and turbulence
in complex terrains. These CFD models could provide opportunities to account for variations
of the flow-field in space and time. However, the computational cost of such a model for
emission inversions will be high, compared to a simple Gaussian approach.

This study found that despite large uncertainties in net emissions estimated using ABC
measurements, these estimates provide some information on the spatial distribution of
emissions within a landfill. The spatial distributions of $CH_4$ emissions from individual source
regions revealed mainly three source areas (A-4 to A-6) contributing to the total estimated
methane emissions from most of the campaigns. The inversions using EF measurements also
identified that the A-6 source area was a significant contributor to net methane emissions, with
additional contributions from A-4 and A-5 and a small proportion of total methane emissions
from the A-1 source area. As the area in the west side of the landfill was covered , it is likely
that methane emissions are significantly reduced, as the covering is designed to improve biogas
capture for electricity production  on-site. Together, these factors may explain the higher
emissions on the eastern side (A-4, A-5, and A-6) of the landfill, relative to the western side
(A-2 and A-3). Measurements with a terrain-resolving flow and dispersion model may provide
better information about the spatial distribution of emission sources within the landfill, as
would more replicate sniffing campaigns similar to those described in this study.

The information about the distribution of the emission sources from the inversions and the
hotspots identified from the sniffing campaigns helps site operators to prioritize mitigation
actions (cover improvement, improvement of landfill gas network collection etc.) . Typically,
at landfill sites, emission sources are highly variable in space and time, with individual sources
within the landfill ranging from sporadic to continuous, and spatially heterogenous hotspots to
large diffusive areas. The analysis of measurements and inversions from the different MGLs
help to provide some qualitative information about the potential emission sources but their
ability to precisely locate the exact spatial distribution of these sources, is limited by the
distance between the vehicle (road) and the sources. Regular sniffing campaigns by foot or by
drone using a portable analyzer and GPS module help, to some extent, to locate certain
suspected hotspots.

## 7    Conclusions

In this study, we present long-term near-surface mobile measurements from 21 campaigns, for
reliable quantification of total methane emission from an active landfill using atmospheric
inversion modelling. We applied a simple inversion approach to quantify methane emissions
from the landfill using a Gaussian plume model. Measurements from a remote EF road, further
away from the landfill, were preferable for inverse modeling as the estimates based on these
measurements were proven to be only weakly sensitive to the defined potential emission
sources and other influencing parameters in the inversions. The total $CH_4$ emissions estimated
using different definitions of potential sources and using data where sampling was conducted



on the distant EF road (11 campaigns) varied from ~0.4 t CH$_4$/d to ~7 t CH$_4$/d, with an average flux value of ~2.1 t CH$_4$/d. These estimated landfill methane emissions showed large temporal variability. Emission estimates based on the measurements conducted along the perimeter of the landfill (ABC) were very sensitive to the characterization of potential emission sources, and were limited in their ability to provide representative landfill emission estimates. An analysis of these measurements helped to provide some insights about potential landfill emission sources, but this information remained insufficient to define the exact spatial distribution of the emission sources within the site. However, targeted sniffing campaigns within the landfill site, identified 9 hotspot emissions sources. Based on our estimated landfill emissions using EF road measurements, we found a weak negative correlation between emissions and atmospheric pressure, and a slightly stronger inverse relationship between emissions and atmospheric temperature. To better characterize such relationships and also for more accurate monitoring of landfill emissions, we suggest that emission estimates should be based on longer-term measurements, ideally, made continuously. In order to better utilize these measurements for landfill emission quantification, especially when sampling close to the landfill, we suggest using a more complex model, such as a CFD model, that can resolve the flow field and turbulence induced by the complex landfill topography.

## 8    Acknowledgments

This work was supported by the Chaire Industrielle Trace ANR-17-CHIN-0004-01 co-funded by the ANR French national research agency, TotalEnergies-Raffinage Chimie, SUEZ, and THALES ALENIA SPACE. We would also like to thank the SUEZ site staff for their assistance in the measurement campaigns. We also acknowledge IFP Energies nouvelles-Géoscience, France for participating to some of the campaigns.

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
