# Peer review of "Detection and long-term quantification of methane emissions from an active landfill"

_Atmospheric Measurement Techniques, 2023_

## Referee Comment (RC1)

**Detection and long-term quantification of methane emissions from an active landfill**

Overall Comments:

This paper uses ground based atmospheric measurement techniques to asses methane emissions from a landfill. Landfills are diverse environments and relatively understudied sources of methane. Given the current global focus on methane this paper offers important and timely information about how to potentially measure emissions from a landfill. While measurements and inversion from MGL are not particularly novel, it is rare that this has been used on a single landfill and with so many measurements and therefore the overall scientific contribution is novel and important.

There are some interesting conclusions reached however I think that the authors could do a better job summarizing and highlighting results that this reviewer thinks are significant. For example, they can identify the spatial sector on the landfill where the majority of emission are coming from (A-6) however they do not highlight this as a finding. This reviewer recommends highlighting that as well as value the ABC road measurements provided in general to understanding of the spatial distribution of emission on the landfill. In addition, more discussion on the activities in each of those spatial sectors would be interesting. They also were able to discover an important aspect of measuring total emission from landfills (you need to be far enough away that your measurement is not significantly influenced by the topography of the landfill, but the increased distance means that the spatial distribution of emissions are difficult to discern.) This is also could be highlighted more.

I believe the methods are well detailed and seem valid but the reviewer is not an expert in inversion and do not feel comfortable giving critical feedback on the details of the inversions.

Overall the paper is well written and organized.

Detailed Comments:

A brief summary of results might be a nice addition to the introduction.

In section 5.1 the authors state some of the most important findings. They use the data from the ABC road to isolate what sectors of the landfill were emitting the most and then they tell us what activities were happening in each sectors. This reviewer thinks this results should be highlighted in the discussion and conclusion. Even if the actual fluxes are uncertain given the topography of the landfill.

Please add a full description in figure 8 so it can be interpreted alone.

Discussion

What activities were happening in A-4, A-5, and A-6 that would cause, what did the sniffer find in these areas. Can you relate the sniffer data, the ABC road, and the EF road data to tell one story about what was causing the methane emissions on this landfill. It seems like you have bits and pieces but it's not consolidated nicely and it is hard for the reader to get what is going on in a holistic way

It would be nice to see some discussion about what might be driving the variability in the emission observed each day from the EF road. For Example, in figure 9 why was Dec 1 so much larger than Feb 5. Is there any data on the activity of the landfill that that might explain this?
.

---

## Referee Comment (RC2)

**Detection and long-term quantification of methane emissions from an active landfill**

This paper presents a long-term study on detecting and quantifying methane emissions from a landfill. The authors conducted 21 measurement campaigns over a period of 3 years, using a combination of ground-based and airborne measurements to estimate methane emissions. They found that the landfill was a significant source of methane emissions, with emissions varying depending on factors such as temperature and precipitation. The study provides valuable insights into reducing greenhouse gas emissions and highlights the importance of long-term monitoring to accurately estimate emissions. This study has more strengths as compared to weaknesses.

**Strengths**

- The study provides a comprehensive analysis of methane emissions from a landfill over a long period of time, which is valuable for understanding the factors that contribute to emissions.
- The authors used a combination of ground-based and airborne measurements, which provides a more complete picture of emissions than using only one type of measurement.
- The study highlights the importance of long-term monitoring to accurately estimate emissions and provides insights into how to improve monitoring methods.

**Weakness**

- The study only focuses on one landfill, so the findings may not be generalizable to other landfills.
- The paper could benefit from more discussion on the implications of the findings for reducing greenhouse gas emissions.

Overall, this paper provides valuable insights into detecting and quantifying methane emissions from landfills and highlights the importance of long-term monitoring for accurately estimating emissions. The study could be improved by providing more discussion on the implications of the findings for reducing greenhouse gas emissions.

---

## Author Comment (AC1)

**Author's response to Anonymous Referee #1**

**Overall Comments:**

This paper uses ground based atmospheric measurement techniques to asses methane emissions from a landfill. Landfills are diverse environments and relatively understudied sources of methane. Given the current global focus on methane this paper offers important and timely information about how to potentially measure emissions from a landfill. While measurements and inversion from MGL are not particularly novel, it is rare that this has been used on a single landfill and with so many measurements and therefore the overall scientific contribution is novel and important.

We thank the reviewer for this general assessment of our manuscript and for the comments, which helped complement our manuscript with some important additions in the abstract, the discussion of the results, and conclusions.

There are some interesting conclusions reached however I think that the authors could do a better job summarizing and highlighting results that this reviewer thinks are significant. For example, they can identify the spatial sector on the landfill where the majority of emission are coming from (A-6) however they do not highlight this as a finding. This reviewer recommends highlighting that as well as value the ABC road measurements provided in general to understanding of the spatial distribution of emission on the landfill. In addition, more discussion on the activities in each of those spatial sectors would be interesting. They also were able to discover an important aspect of measuring total emission from landfills (you need to be far enough away that your measurement is not significantly influenced by the topography of the landfill, but the increased distance means that the spatial distribution of emissions are difficult to discern. This is also could be highlighted more.

I believe the methods are well detailed and seem valid but the reviewer is not an expert in inversion and do not feel comfortable giving critical feedback on the details of the inversions.

Overall the paper is well written and organized.

We thank the reviewer for stressing the need to better highlight these insights. As suggested, we now include additional discussions and emphasize better these conclusions in the revised manuscript:

- We now better highlight the results about the sector A-6 which is one of the major emitting area in the landfill. In this particular area source, there were juncture of biogas pipes networks, wells, leachate ponds, etc. As highlighted by the analysis of near-surface sniffing measurements, many of the identified hotspots corresponded to these sources which likely represent the major part of the methane emissions from sector A-6. We observe significant temporal variability in the emissions from this area (Figure S2.16) and this variation underscores that the elevated emissions primarily coincide with instances of sporadic leakages in the infrastructure of the landfill.

- Measurements taken in proximity to the landfill on ABC road are useful for capturing the spatial distribution of potential methane emission sources within the landfill. However, it is important to note that our use of a simple Gaussian plume model, which doesn't account for topography, in the atmospheric inverse modelling of the methane emissions renders the estimates from near-landfill measurements sensitive to various factors, and therefore, less robust than those from more distant roads.

- therefore, there is a trade-off between obtaining reliable estimates of the total methane emissions and discerning the spatial distribution of the emissions within the landfill when using both close and remote measurements.

- Our information about the day-to-day activity data, particularly for the days around the campaigns, is limited for this landfill. This limitation hampers a more in-depth discussion on the link between the timeseries of the estimated methane emissions and the ongoing activities in each emitting sector within the landfill. However, we discussed the detailed map of the pipes network, wells, leachate ponds, biogas valorization plant, etc., in the landfill, and we used this information to discuss the methane hotspots identified with the near-surface sniffing measurement campaigns. We had highlighted that these identified methane hotspots correspond to potentially emitting structures in the landfill map.

**Detailed Comments:**

A brief summary of results might be a nice addition to the introduction.

Thank you for your suggestion. While we appreciate the idea of including a brief summary of results in the introduction, we feel that the editors, reviewers and readers tend to prefer the introduction not to display such summaries, which are already included in the abstract and conclusion. So we prefer to follow our traditional approach to limit the scope of the introduction to the background and motivation, reserving the summary for the conclusion.

In section 5.1 the authors state some of the most important findings. They use the data from the ABC road to isolate what sectors of the landfill were emitting the most and then they tell us what activities were happening in each sectors. This reviewer thinks this results should be highlighted in the discussion and conclusion. Even if the actual fluxes are uncertain given the topography of the landfill.

As mentioned before, we now better highlight these results based on the ABC measurements in the abstract, discussion and conclusion, with a cautionary note about the uncertainties in the corresponding flux estimates.

Please add a full description in figure 8 so it can be interpreted alone.

We have now added a full description in the caption of figure 8 in the revised manuscript.

**Discussion**

What activities were happening in A-4, A-5, and A-6 that would cause, what did the sniffer find in these areas. Can you relate the sniffer data, the ABC road, and the EF road data to tell one story about what was causing the methane emissions on this landfill. It seems like you have bits and pieces but it's not consolidated nicely and it is hard for the reader to get what is going on in a holistic way

It would be nice to see some discussion about what might be driving the variability in the emission observed each day from the EF road. For Example, in figure 9 why was Dec 1 so much larger than Feb 5. Is there any data on the activity of the landfill that that might explain this?

As discussed in the manuscript, the sectors within the landfill were filled one by one, with A-3, A-6, A-5 and A-4 being the last to be filled (in this order) and, therefore, being the main emission areas.

1) A-4 was the last sector where waste reception was ongoing during this study, particularly in the last phase of the campaigns when we had reliable onsite meteorological measurements to support the analysis of the emission spatial distribution within the landfill using the ABC measurements. During active waste reception, the corresponding reception areas, here A-4, was open and uncovered, and A-4 was thus restricted from any sniffing measurements due to safety considerations. However, the analyses with the ABC measurements and with the inversions identified A-4 as the most continuous emitter, and as the largest source area on average, which is consistent with the fact that it remained active, uncovered and open throughout the campaigns.

2) Other sectors A-5, A-6, and A-3 were active during mainly initial phases of the measurements campaigns. These source areas were covered with clay and membrane during most of the campaigns, but there were juncture of biogas network purges, biogas network wells, bioreactor tank, etc. on these areas. As already discussed in the section 3.1, the main hotspots identified from the near-surface sniffing measurements mainly correspond to the leaks in theses landfill infrastructures. Many of these identified hotspots were located in the A-5 and A-6 landfill areas which explains the higher emissions from these source areas. As already discussed in the manuscript, the EF measurements appear to be too far from the landfill, to support the assessment of the distribution of the sources within the landfill.
These points are now better highlighted in the revised manuscript.

3) As already highlighted in Section 6, the estimates of the total methane landfill emissions based on the EF measurements exhibit significant temporal variability, even over a short period of a few days. For instance, the total methane emissions on December

08, 2020 (~1.25 t CH4/d) were considerably smaller than those on December 01, 2020 (~7 t CH4/d), despite a one-week interval between these two sampling campaigns and despite the fact that both measurement campaigns were conducted during the similar daytime hours. We attempted to analyze the variability in estimated emissions using atmospheric pressure and temperature data obtained from a nearby meteorological station. However, we found only a weak correlation of the emission estimates with these variables. We anticipate that the high temporal variability in the emissions is primarily attributable to landfill activity, such as the fixing of a large methane leak, which can lead to a substantial drop in emissions within a short timeframe. However, as we mentioned previously in reply to another comment, the limited availability of day-to-day activity data for this landfill make it challenging to relate such variability in our estimates based on the EF measurements with the landfill activities.

---

## Author Comment (AC2)

**Author's response to Anonymous Referee #2**

This paper presents a long-term study on detecting and quantifying methane emissions from a landfill. The authors conducted 21 measurement campaigns over a period of 3 years, using a combination of ground-based and airborne measurements to estimate methane emissions. They found that the landfill was a significant source of methane emissions, with emissions varying depending on factors such as temperature and precipitation. The study provides valuable insights into reducing greenhouse gas emissions and highlights the importance of long-term monitoring to accurately estimate emissions. This study has more strengths as compared to weaknesses.

We thank the reviewer for this general assessment of our manuscript and for the comments, which helped complement our manuscript with some additions in the introduction and discussion of results.

**Strengths**
- The study provides a comprehensive analysis of methane emissions from a landfill over a long period of time, which is valuable for understanding the factors that contribute to emissions.
- The authors used a combination of ground-based and airborne measurements, which provides a more complete picture of emissions than using only one type of measurement.
- The study highlights the importance of long-term monitoring to accurately estimate emissions and provides insights into how to improve monitoring methods.

We thank the reviewer for listing the strengths of the study. We would just like to indicate that the measurements conducted within the landfill are on foot measurements while "airborne measurements" is rather used for measurements from drones or aircrafts which is why we do not use such a term in the manuscript.

**Weakness**
- The study only focuses on one landfill, so the findings may not be generalizable to other landfills.

The landfills differ in terms of topographical features, types of wastes, management practices by the site operators, etc. Therefore, we will need other studies such as this one to support the establishment of standard atmospheric monitoring techniques that are robust for any landfills, and even more studies to improve the definition of emission factors associated to the landfill methane emissions in general.

We needed much resources and efforts for a long-term monitoring of the emission from the landfill we study here. Some studies cover several landfills (e.g., Mønster et al., 2015) but with two drawbacks:

- by providing single or few instant emissions estimates

- by using a monitoring approach for a given landfill which is often limited by the lack of specific measurements or of analysis due to the need for a relatively fast approach when covering many sites

Both types of studies are complementary, and the build-up and progress of the two types of studies should support the development and improvement of the atmospheric monitoring techniques and the derivation of more accurate emission factors and understanding of the process governing the landfill emissions. Our study participates to the corresponding long-term effort, but it could not have the too ambitious objective of providing a general conclusion on the landfill emissions in general. However, our atmospheric measurement strategy and the inversion approach to estimate the methane emission is general enough to be followed to monitor the emissions from other landfills. Our methodological conclusions can be used as a basis for future applications and developments.

We now better clarify this point in the introduction and discussion of the manuscript.

- The paper could benefit from more discussion on the implications of the findings for reducing greenhouse gas emissions.

   This connects to the previous comment: as indicated above, we now clarify in the introduction and discussion the scope of our conclusions. We also remind the need for better methane emission factors to support the reduction of greenhouse gas emissions.

Overall, this paper provides valuable insights into detecting and quantifying methane emissions from landfills and highlights the importance of long-term monitoring for accurately estimating emissions. The study could be improved by providing more discussion on the implications of the findings for reducing greenhouse gas emissions.

We thank the reviewer for these final remarks and as detailed above, we have included some additional discussions in the revised manuscript following these recommendations.

**References**

Mønster, Jacob, et al. "Quantification of methane emissions from 15 Danish landfills using the mobile tracer dispersion method." *Waste Management* 35 (2015): 177-186.